# Heart Rate Responses of Post-Menopausal Women to Zumba Gold^®^ Classes

**DOI:** 10.3390/biology13070462

**Published:** 2024-06-21

**Authors:** Anne Delextrat, Clare Dorothy Shaw, Alba Solera-Sanchez

**Affiliations:** Department of Sport and Health Sciences and Social Work, Oxford Brookes University, Oxford OX3 8HU, UK; clareshaw@brookes.ac.uk (C.D.S.); asolera-sanchez@brookes.ac.uk (A.S.-S.)

**Keywords:** moderate and vigorous physical activity, age, body fat, health

## Abstract

**Simple Summary:**

The aim of this study was to observe the heart rate (HR) of post-menopausal women during Zumba Gold^®^ classes, and to determine if these rates vary with body fat. Twenty-three post-menopausal women (68.8 ± 7.2 years old; 160.0 ± 5.2 cm; 66.9 ± 11.1 kg, 36.0 ± 9.9% body fat) participated. After measuring their height, weight, body fat and fitness, HR measurements were taken during four of their regular Zumba Gold^®^ classes. The average HR (HR_mean_), as well as time spent in different HR intensity categories, was calculated. Our results showed an average HR of 70.2% of the maximal HR. Women with lower body fat achieved a higher HR_mean_ and spent less time at light to very light intensity and more time at moderate intensity compared to those with higher body fat. These results suggest that Zumba Gold^®^ can be an effective exercise option for post-menopausal women aiming to meet the recommended daily exercise guidelines, with leaner women expending more energy than women with a greater body fat percentage.

**Abstract:**

This study aimed to describe the heart rate (HR) responses of post-menopausal women during Zumba Gold^®^ classes and to investigate the effects of body fat on HR responses. Twenty-three post-menopausal women (68.8 ± 7.2 years old; 160.0 ± 5.2 cm; 66.9 ± 11.1 kg, 36.0 ± 9.9% body fat) participated. Baseline testing assessed participants’ anthropometric and fitness characteristics. Then, HR measurements were taken during four of their regular Zumba Gold^®^ classes, and average HR (HR_mean_), as well as time spent in different HR intensity categories, was calculated. Linear regressions and *t*-tests were performed to analyse the data. The average HR during Zumba Gold^®^ classes was 70.2% of maximum HR. Women with lower body fat achieved a significantly higher HR_mean_ and spent less time at light to very light intensity and more time at moderate intensity compared to those with higher body fat. Body fat percentage and age were identified as determinants of time spent at moderate intensity. These findings suggest that Zumba Gold^®^ can be an effective exercise option for post-menopausal women aiming to meet the recommended daily exercise guidelines. Understanding the HR responses during Zumba Gold^®^ classes can aid in the development of safe and effective exercise prescriptions for this population.

## 1. Introduction

Menopause is associated with various physical and physiological changes, such as sarcopenia, increased fat mass and its redistribution [1,2]. These changes can drastically affect women’s health and lifestyle, as they can lead to obesity, osteoporosis, a greater risk of cardiovascular disease, depression and an overall decline in functional capacity [1,3].

Physical exercise has many well-established benefits that prevent, delay or reverse these physiological changes in this population [4,5,6]. However, menopause is also a time when a large number of women decrease their physical activity (PA) levels [7], and this was aggravated by the COVID-19 pandemic lockdown that resulted in increased sedentariness [8]. Zumba has been reported in the top six favourite physical activities of middle-aged women [9] and, hence, seems like an attractive option to promote a healthier lifestyle. Zumba Gold^®^ is recommended for those over 50 years old because it has lower-impact movements. In contrast to the growing worldwide participation in Zumba Gold^®^, there is very limited evidence about the physiological responses during a workout. It is essential to precisely quantify cardiovascular and respiratory responses to exercise to be able to safely prescribe PA and rehabilitation programmes for post-menopausal women.

To our knowledge, only two studies have investigated the acute responses to Zumba Gold^®^ specifically in post-menopausal women. Dalleck et al. [10] reported a mean heart rate (HR) of 50.3% of HR reserve during a session, characteristic of moderate intensity in women aged 63 years old. Exercising at moderate intensity has also been recommended to improve/maintain physical health and prevent cardiovascular disease [11]. However, this study tested eight female participants during only one Zumba Gold^®^ class, and therefore additional data on a larger sample size and over several sessions are needed to confirm these results. Rossmeissl et al. [12] recorded HR data during three Zumba Gold^®^ classes at weeks 1, 5 and 12 of an intervention programme and showed median HRs of 69%, 75% and 72% of HR_max_. In addition, they reported that less than 5% of the session time was spent with a HR lower than 60% of HR_max_, and that women spent 26–47%, 40–50% and 5–21% of the time in HR zones that were 60–69%, 70–79% and 80–89% of HR_max_, respectively. With regards to the implications for health in this population, this study also showed that Zumba Gold^®^ classes increase quality of life and decrease menopausal symptoms [12]. The sample size in the aforementioned study was relatively small (*n* = 11 to 14), and the women tested were also overweight, which somewhat biases the applicability of these findings to all post-menopausal women. Indeed, it has been shown that healthy-weight adult women spent a significantly lower percentage of Zumba class time at the lower levels of intensity (sedentary and lifestyle activity levels) and significantly more time at the highest levels (vigorous and very vigorous) compared to overweight and obese women [13]. The percentage of time at moderate and vigorous intensities in this study was 62.1 ± 15%, 50.1 ± 9.4% and 44.1 ± 11.9% in healthy-weight, overweight and obese women, respectively [13].

The fact that leaner women seem to spend more time at higher HR levels than women with greater body fat is very important to highlight, because one of the main marketing strategies of the Zumba^®^ Fitness industry is around weight loss. It is crucial to inform women that, based on the results previously described [13], they may not be able to work as hard and, hence, lose as much weight as they first believed [14] if they are overweight, for example. It should be noted, however, that these results [13] need to be confirmed by further studies. Similarly, there might be other determinants of HR responses during Zumba Gold^®^ classes that could be important to highlight to consumers. For example, older adults with a better aerobic capacity have been shown to reach significantly greater HR levels during physical exercise [15]. In addition, it was suggested that individuals with reduced balance may not be able reach high levels of HR during dance-based exercise sessions because some dance steps (in particular backward steps) challenged their balance [16].

Within this context, the aim of the present study was to describe the HR responses of post-menopausal women during Zumba Gold^®^ classes. Secondary aims were to identify the determinants of these responses and to investigate the effects of body fat on HR responses.

## 2. Materials and Methods

### 2.1. Participants

Twenty-three post-menopausal women (68.8 ± 7.2 years old; 160.0 ± 5.2 cm; 66.9 ± 11.1 kg, 36.0 ± 9.9% body fat) volunteered to take part in this study. They were recruited from existing Zumba Gold^®^ classes in the community. Inclusion criteria were familiarity with the practice of Zumba Gold^®^ for a minimum of 3 months prior to the study and no injury at the time of the study. Post-menopausal was defined as an entire year without bleeding (www.nhs.uk, accessed on 7 June 2024). They were further divided into healthy-weight (n = 11; 68.0 ± 8.5 years old; 160.6 ± 4.8 cm; 59.7 ± 5.0 kg, 29.2 ± 9.3% body fat) and overweight and obese (n = 12; 69.5 ± 6.0 years old; 159.7 ± 5.7 cm; 73.4 ± 11.2 kg, 42.3 ± 5.2% body fat) groups. These groups were based on the body mass index criteria such that anyone over 25 kg·m^−2^ was considered overweight and anyone over 30 kg·m^−2^ was considered obese, as defined by the World Health Organization (www.who.int, accessed on 7 June 2024). Participants’ experience with Zumba Gold^®^ was 1.6 ± 0.8 years, and at the time of the study, they were involved in one to two weekly Zumba Gold^®^ classes (1.2 ± 0.4 weekly sessions). Each participant was informed in detail about the testing procedures and possible risks of the study before signing an informed consent in line with the approval by the local ethics committee (approval number 201495).

### 2.2. Procedures

After a baseline testing session to check eligibility criteria and collect anthropometric and fitness characteristics, HR measurements were taken during participants’ usual weekly Zumba Gold^®^ classes in the community. A total of six weekly classes led by four different instructors were used to capture the variety of practices between instructors and avoid the bias of a single class approach. Each participant had their HR measured during four different classes so that we could obtain a more representative idea of their physiological responses.

### 2.3. Baseline Testing

Height (m) was measured with a portable scale (Seca, Marsden, UK), while body mass (kg), body fat (BF, %) and muscle mass (kg) were measured by bioelectrical impedance using the Tanita BC 418MA Segmental Body Composition Analyser (Tanita Corporation, Tokyo, Japan). Participants were required to avoid consuming caffeine 24 h before the measurements and avoid any vigorous activity 8 h before the measurements.

The following fitness characteristics were measured to investigate if they were determinants of HR responses during the classes:

The 6 min walk test was used to measure participants’ cardiorespiratory fitness. It consisted of walking as fast as possible between two cones placed 10 m apart for six minutes. The total distance covered was measured. This test is characterised by good validity and reliability (test–retest correlation coefficient of r = 0.88) in older adults [17].

Lower body muscular strength was assessed using the 30 s chair stand. It is part of the Senior Fitness Test (SFT) battery [17] and consists of sitting on a 17-inch-high chair with arms crossed on the chest, standing up fully and returning to a fully seated position as many times as possible for 30 s. This test is reliable (test–retest correlation coefficient of r = 0.89 in older women) and characterised by good criterion validity [17].

The lower quarter Y-balance test was used to measure dynamic balance. It consists of maintaining balance while standing on one leg, with the contralateral leg reaching in three different directions (anterior, posteromedial and posterolateral). The test was repeated twice, and distance reached in each of the three directions was measured and the sum calculated as the main performance outcome. Freund et al. [18] reported good reliability for the mean reach distances in each direction (intraclass correlation coefficients ranging from 0.94 to 0.99).

### 2.4. Zumba Gold^®^ Sessions

HR measurements were taken biweekly during participants’ regular Zumba Gold^®^ classes in the community. Each Zumba Gold^®^ class was 60 min long and always included the same songs and dance moves to make sure participants were equally familiar with the class. Classes were led by a qualified instructor who was certified by the Zumba^®^ Fitness company and were performed indoors in dance studios. Before each song, the instructor demonstrated the steps slowly, giving time for participants to learn them. Each session included one or two warm-up and cool-down songs, and the main body was based on steps from the following six dance styles commonly used in Zumba: merengue, cumbia, reggaeton, salsa, belly dancing and pop. Movements based on these songs differed in their intensity and the type of steps involved. Participants were required to arrive to the class early and were fitted with a HR monitor (Polar H7, Polar, Kempele, Finland). Participants’ HRs were then recorded at 1 s intervals during Zumba Gold^®^ classes. A total of four HR recordings were obtained for each participant over a period of 8 weeks. Four classes were chosen to avoid bias in comparisons linked to the intrinsic characteristics of one single class and variability in participants’ performance.

From the recordings, the average HR (HR_mean_) was calculated as the average of HR values from the start of the warm-up period to the start of the cool-down period. The cool-down period was excluded because it consisted mostly of static stretching exercises (very low intensity), while the warm-up period was a dance routine similar to the main body of the class. HR values were expressed as absolute (beat min^−1^) and relative (% of maximal HR (HR_max_)), according to the Tanaka equation [19]. Finally, the time spent in the following HR intensity categories was calculated [11]:-Very light to light: HR < 64% of HR_max_-Moderate: 64% of HR_max_ ≤ HR ≤ 76% of HR_max_-Vigorous to maximal: 76% of HR_max_ ≤ HR ≤ 95% of HR_max_-Near maximal: HR > 95% of HR_max_

### 2.5. Statistical Analyses

All data were expressed as mean ± standard deviation, with 95% confidence intervals. Normality was checked by the Shapiro–Wilks test. A linear regression was performed to identify the determinants of HR_mean_ and the time spent in each HR intensity category. In addition, to compare the data measured in the healthy-weight and overweight/obese group, a Student’s *t*-test for independent samples was performed. Effect sizes were calculated using Cohen’s d, with the values of 0.2, 0.5 and 0.8 interpreted as small, moderate and large, respectively [20].

## 3. Results

Performance and HR data for all participants, as well as for each group, are presented in Table 1. The average duration of the Zumba Gold^®^ classes, excluding the cool-down periods, was 54.8 ± 1.6 min.

The linear regression showed that body fat percentage accounted for 22.2% of the variance in the time spent at very light to light intensity (Table 2). There were two significant predictors of the time spent at moderate intensity, body fat percentage and age, that together accounted for 35.1% of the variance in this variable. (Table 2). However, linear regression showed that there were no significant predictors of the HR_mean_ measured during the Zumba Gold^®^ classes, as well as the percentage of time spent at vigorous and near maximal intensities.

There were significant differences between weight groups in weekly PA (*p* = 0.045, d = 1.1), body fat percentage (*p* < 0.001, d = 1.77), muscle mass (*p* < 0.001, d = 1.65), HR_mean_ (*p* = 0.047, d = 0.72) and the percentage of time spent at very light to light intensity (*p* = 0.010, d = 1.1) and moderate intensity (*p* = 0.045, d = 0.74). No significant difference between groups was observed for any other parameter (*p* > 0.05).

## 4. Discussion

The main findings of the present study show an average HR during Zumba Gold^®^ of 70.2% of HR_max_ in post-menopausal women. In addition, women with less body fat achieved a significantly greater HR_mean_ and spent significantly less time at very light to light intensity and significantly more time at moderate intensity compared to women with greater levels of body fat. Finally, the main determinants of the time spent at moderate intensity were body fat and age.

The average HR measured during the Zumba Gold^®^ classes of the present study is in line with previous studies. Indeed, Dalleck et al. [10] found that the HR_mean_ corresponded to 68.7% of HR_max_ in nine healthy-weight (BMI of 22.6 kg·m^−2^) women aged 63.0 years. In addition, Rossmeissl et al. [12] reported median HRs of 69%, 75% and 72% of HR_max_ during three Zumba Gold^®^ sessions in post-menopausal overweight women. The slightly lower HR in the present study may be attributed to the higher body fat percentage of our participants (42.3 ± 5.2% vs. 39 ± 8%, respectively, in the present study and in the study of Rossmeissl et al. [12]). While it is well established that individuals with a greater percentage of body fat have more difficulty performing weight-bearing exercises due to the extra fat mass they are carrying (Primavesi et al. [21]), our results could also be due to other factors. Indeed, it was suggested that participants’ experience with Zumba could influence their familiarisation with the steps and, hence, allow them to perform at a greater intensity [13]. Our participants had an average of 2.6 ± 2.0 years of experience with Zumba Gold^®^, which suggests good familiarisation; however, this cannot be compared with the studies of Dalleck et al. [10] and Rossmeissl et al. [12], as they did not provide this information. Other factors that could influence HR responses but were not measured in previous studies (or ours) are music tempo and air temperature.

A comparison with other dance styles shows that Zumba Gold^®^ induces a greater HR compared to several other types of aerobic exercise, such as walking (60 ± 8% of HR_max_ in women aged 68 years [22]) or other dance types, including cultural, ballroom and classical (50–70% of HR_max_ in a meta-analysis by Rodriguez-Krause et al. [23] that included women aged 58–72 years). On the other hand, the average HR in our study is slightly lower than values observed during aerobic dance (74 ± 12% of HR_max_ in women aged 68 years [22]) and Scottish dance (72 ± 7% of HR_max_ in post-menopausal women aged 50–80 years [24]). Therefore, Zumba Gold^®^ could be considered to be in the top range of aerobic exercises in terms of exercise intensity that post-menopausal women could choose from.

The recommendations from the American College of Sports Medicine (ACSM) for older adults are to spend 30 min daily performing moderate intensity exercise. Our results suggest that Zumba Gold^®^ can help achieve this goal, as 74.5% of the Zumba Gold^®^ classes was spent at moderate intensity or above, which corresponds to 40.8 min. The benefits of exercising at moderate intensity or higher are well established. Indeed, moderate intensity exercise utilises and enhances the ability of skeletal muscles to oxidise fatty acids, improves mitochondrial density and leads to improvements in aerobic capacity [25]. Moderate to high intensity exercise promotes cardiac hypertrophy and thus improves stroke volume and cardiac output [26]. These adaptations should, therefore, influence menopausal women’s health in the long term by preventing or delaying cardiovascular disease and promoting weight loss or maintenance, which are known health issues associated with menopause [1,2]. To our knowledge, the only study reporting HR intensity categories during Zumba Gold^®^ shows that participants spent more than 95% of the class above 60% of HR_max_ and 45–71% of the class above 70% of HR_max_, which is greater than in our study [12]. These discrepancies could be explained by the higher body fat percentage of the participants in our study, as previously explained, and possibly by their older age (68.8 ± 7.2 vs. 55 ± 6 years). While the physiological effects of aging on cardiovascular function are well established [27], the results from linear regression in our study indicated that age was a significant predictor of the time spent at moderate intensity during Zumba Gold^®^. Post-menopausal women and exercise instructors should therefore be aware of these variations in relation to age (and body fat) to manage their and their clients’ expectations. Nevertheless, low intensity exercise improves circulation and increases muscular capillary beds and oxygen delivery [28], which could also contribute to some health benefits around menopause, such as increasing aerobic capacity. Finally, another variable to consider in relation to age is musculoskeletal (MSK) conditions, which could prevent women from reaching higher heart rates. For example, in our sample, three women in each group reported a long-term MSK condition (not caused by Zumba) affecting their back (n = 1), hip (n = 2), knee (n = 1) or foot (n = 2).

An interesting result of the present study is that women with less body fat (healthy-weight group) had a significantly greater HR_mean_ and spent significantly less time at light to very light intensity and significantly more time at moderate intensity compared to the overweight and obese group. A study in younger women found a similar result during Zumba Fitness classes, with the healthy-weight group spending significantly more time in the moderate and vigorous PA (MVPA) category compared to an overweight group (62.1 ± 9.4% vs. 50.1 ± 9.4% of the session time) [13]. These authors suggested that the greater exercise intensity observed in the healthy-weight group could be partly attributed to a greater muscle mass and/or greater PA levels. Our results support this hypothesis, since muscle mass was significantly greater in the healthy-weight group compared to the overweight and obese group. Additionally, although balance did not differ between groups, there were significant differences in the amount of weekly PA, with the healthy-weight group engaging in more PA. These findings are particularly useful to inform women when they choose an exercise class, especially if their goal is weight loss, as Zumba Gold^®^ might not result in enough energy expenditure for overweight and obese women to achieve this goal. Finally, and in relation to this last point, we observed large variability between participants’ HR values and time spent in various HR intensity categories. This is an important characteristic of Zumba Gold^®^ and the Zumba culture in general, as the class does not adopt a strict choreography, and instructors advise attendees to roughly follow or create their own dance moves instead. Consequently, post-menopausal women taking part in Zumba Gold^®^ could experience a variety of benefits specific to each HR intensity category, such as fat burning or improvement in aerobic fitness [29].

The main limitation of this study is the lack of data collected on energy expenditure and cardio-respiratory data that could have allowed us to precisely quantify energy expenditure during the classes.

## 5. Conclusions

In conclusion, our results showed that Zumba Gold^®^ allows for an exercise intensity that is high enough to address the guidelines for PA in post-menopausal women. In addition, the HR reached during the class depends on body fat and age to a lesser extent. These findings are important to inform exercise choice for post-menopausal women, whether their aim is weight loss, general fitness or maintenance and improvement of muscle mass. With consideration of the recommendation from the ACSM to spend 30 min per day performing moderate intensity exercise alongside this study’s findings, both aerobic fitness and fatty acid oxidation capacity can be improved through Zumba Gold^®^, which can, therefore, help to mitigate some physiological effects of menopause. With adaptations to fat metabolism, which aids weight management and improves cardiovascular health, it can be concluded that moderate or moderate to high intensity zones are beneficial for menopausal women. Further studies should investigate the physiological and psychological benefits of Zumba Gold^®^ in post-menopausal women.

## Figures and Tables

**Table 1 biology-13-00462-t001:** Exercise habits, performance and heart rate (HR) data for all participants and each group, expressed as mean ± standard deviation and 95% confidence intervals for the difference (HR_mean_: average HR; 6MWT: 6 min walk test).

Variable	All (n = 23)	Healthy Weight (n = 11)	Overweight/Obese (n = 12)	95% CI
Zumba experience (years)	1.6 ± 0.8	1.8 ± 0.9	1.3 ± 0.7	−0.6 to 1.5
Weekly PA (hours)	2.6 ± 1.5	3.3 ± 1.4	1.8 ± 1.3 *	−3.0 to 3.4
Body fat (%)	36.0 ± 9.9	29.2 ± 9.3	42.3 ± 5.2 **	−19.6 to −6.7
Muscle mass (kg)	26.9 ± 4.3	29.8 ± 4.4	24.3 ± 2.0 **	2.6 to 8.4
HR_mean_ (% of HR_max_)	70.2 ± 6.0	72.4 ± 3.8	68.2 ± 7.1 *	−0.8 to 9.2
Very light to light intensity (% time)	25.5 ± 24.7	13.3 ± 11.8	36.7 ± 28.5 *	−42.5 to −4.3
Moderate intensity (% time)	48.0 ± 19.3	55.1 ± 16.7	41.5 ± 19.9 *	−2.2 to 29.6
Vigorous to maximal (% time)	25.6 ± 20.1	30.1 ± 13.0	21.5 ± 24.8	−8.6 to 25.8
Near maximal (% time)	0.9 ± 3.1	1.4 ± 4.4	0.4 ± 1.1	−1.7 to 3.8
6MWT (m)	477 ± 124	495 ± 90	460 ± 151	−74 to 144
Sit and stand (repetitions)	15.2 ± 3.9	14.4 ± 4.0	16.0 ± 3.9	−5.1 to 1.8
Y balance right (cm)	81.7 ± 13.7	85.7 ± 16.3	77.9 ± 10.0	−3.9 to 19.4
Y balance left (cm)	83.2 ± 13.6	85.0 ± 15.4	81.5 ± 12.2	−8.5 to 15.5

PA: physical activity; HR: heart rate; 6MWT: 6 min walk test. *: significant difference between groups, *p* < 0.05. **: significant difference between groups, *p* < 0.001.

**Table 2 biology-13-00462-t002:** Linear regression statistics for the prediction of the time spent at various exercise intensities.

Variable	r^2^ for the Model	Beta	t	*p*
Predictors of the time spent at very light to light intensity				
Body fat percentage	0.222	0.471	2.447	0.023
Predictors of the time spent at moderate intensity				
Age	0.351	0.441	2.445	0.024
Body fat percentage	0.351	0.422	2.335	0.030

## Data Availability

Data supporting reported results can be found on the main author’s Google Drive: https://drive.google.com/drive/folders/1ZOlPskYcTfbZFV-ex5ZsDAKAlxwEEpGH, accessed on 31 May 2024.

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
