# Peer review of "Heart Rate Responses of Post-Menopausal Women to Zumba Gold® Classes"

_biology, 2024, doi:10.3390/biology13070462_

Round 1
Reviewer 1 Report
Comments and Suggestions for Authors
This article analyses the heart rate variability of a group of women performing a specific physical activity. The article is well presented, although it does not add much new data to the scientific literature.
The inclusion of the physical condition tests in the methodology and results table is not well understood and then it does not go into the analysis of these with heart rate variability. It is recommended that this assessment be taken into account.
Author Response
Reviewer 1
Overall comment
This article analyses the heart rate variability of a group of women performing a specific physical activity. The article is well presented, although it does not add much new data to the scientific literature.
Response
Thank you for your comments. Nevertheless, we believe that having a larger sample of participants as well as longer data collection timepoints than previous peers, could make the data presented highly valuable for further applied research and practice in this population. The inclusion of a regression analysis including fitness elements is novel, as no study has investigated this before. Also, the effect of body fat on HR responses was only shown by one research team, and hence need to be confirmed by more studies.
Comment 1
The inclusion of the physical condition tests in the methodology and results table is not well understood and then it does not go into the analysis of these with heart rate variability. It is recommended that this assessment be taken into account.
Response
The aim of this inclusion is to try to understand what are the fitness determinants of HR responses during Zumba Gold, to better inform women taking part in these classes, so that they don’t have the wrong expectation. It was taken into account in our regression analysis, but we agree that we did not explain or justify our aim very well. The follwoing paragraph was added at the end of the introduction to clarify this (as a result, 3 references were added):
“The fact that leaner women seem to spend more time in intense HR zones than women with greater body fat is very important to highlight because one of the main marketing strategies of the Zumba® Fitness industry is around weight loss. It is crucial to inform women that, based on the results previously described [13], they may not be able to work as hard, and hence lose as much weight as they first believed [14] if they are overweight, for example. It should be noted, however, that these results [13] need to be confirmed by further studies. Similarly, there might be other determinants of HR responses during Zumba Gold® classes that could be important to highlight to consumers. For example, older adults with a better aerobic capacity have been shown to reach significantly greater HR levels during physical exercise [15]. In addition, it was suggested that individuals with reduced balance may not be able reach high levels of HR during dance-based exercise sessions, because some dance steps (in particular backward steps) challenged their balance [16].”
Comment 2
The authors collected valuable and relevant data that can be used in applied practice. However, there are some points that should be improved upon and clarified.
Response
We appreciate your suggestions. We have made amendments in the paper and hope it is now clearer.
Reviewer 2 Report
Comments and Suggestions for Authors
The authors collected valuable and relevant data that can be used in applied practice. However, there are some points that should be improved upon and clarified.
Introduction
Line 25: "The" in "The menopause" should be removed and should read as "Menopause".
Line 26 - 27: "Decline in psychological and cognitive function and mood swings?" should either be removed or made relevant to the outcome measures and research question of this study. These items were not measured. However, the authors could relate the HR measures to health including cognitive and mood health. This should be done in the Introduction as well as the Discussion. Or should be removed.
Line 31: "Psychological changes"; same as above comments. Must be made relevant to outcome measures and research question or removed.
Lines 41 - 60: The implications and relevance of the cited data on other Zumba HR studies should be made more clear. Such as the implications on health and post-menopausal symptoms, etc.
Methods
Participants: Zumba experience and frequency should be reported here.
Line 73 - 75: What were the classification ranges for body fat categories? And who/what were their origin?
Baseline Testing: No rationale was given for fitness testing? Were these used to characterize the physical fitness/capacity of participants? If so, this is highly unclear. This needs to be explained in the Methods as well as discussed in the Discussion. Without a rationale for these tests, the reader does not understand the reason, relevance, and implications of them.
Line 87 - 88: Were pre-test procedures followed for BIA testing? If so, state them. Additionally, muscle mass was recorded; that should be stated here as well.
Line 108: Define what the criteria used for a "qualified" instructor.
Line 108 - 109: Were all subjects equally familiar or unfamiliar with the songs, moves, etc. used in the sessions? Were song speeds similar? Were sessions, songs, moves of similar intensity?
Line 124: I do not believe ACSM uses the "Zone" terminology (i.e. Zone 1, Zone 2, etc.). If citing ACSM, use to cite %HRmax intensity classifications (i.e. light, moderate, high-intensity). If using "Zone" terminology, cite reference of where that comes from.
Results
Table 1:
- n for All and each weight category needs to be included.
- Zumba experience and Weekly PA units; is this weekly h? "h" is insufficient.
- Up to Zone 2 row, text is bold, after text is not bold. Needs to be consistent.
- All acronyms must be defined under table.
Line 151: "However" needs a space before previous sentence.
Line 146 - 158: These data need to be depicted in table of figure.
Line 154 - 156: "Groups" need to be clearly defined and stated (i.e. weight groups").
Discussion
The paper begins with discussing health issues related to menopause, however, these are not discussed after that. The Discussion should tie back to those opening comments and discuss the implications that these data have on post-menopausal women's health. The Discussion currently does discuss the significance, implications, or applications of the data. The authors should include a discussion on these items to justify the study and to highlight it's relevance.
Line 178: Years of experience should be in Table 1.
Comments on the Quality of English Language
Overall, clear English writing.
Author Response
Reviewer 2
We believe that we have addressed all your comments. We were just a bit confused with the level of written English. You ticked a box stating that the standard of English needed improvement but later in the text they mentioned that the article was very well written. One of the co-authors is a native english speaker and has reviewed the article before our submission, but there wasnt much to correct. Hope this is ok.
Comment 1: Introduction
Line 25: "The" in "The menopause" should be removed and should read as "Menopause".
Response: done
Line 26 - 27: "Decline in psychological and cognitive function and mood swings?" should either be removed or made relevant to the outcome measures and research question of this study. These items were not measured. However, the authors could relate the HR measures to health including cognitive and mood health. This should be done in the Introduction as well as the Discussion. Or should be removed.
Response: we have removed this too.
Lines 41 - 60: The implications and relevance of the cited data on other Zumba HR studies should be made more clear. Such as the implications on health and post-menopausal symptoms, etc.
Response
Thanks for your suggestions. We have added the benefits of Zumba Gold on Quality of Life and Menopausal symptoms as reported in the study of Rossmeissl et al (the Dalleck study did not include any benefits), see below:
“Line 66-67: With regards to the implications for health in this population, this study also shows that Zumba Gold® classes increase quality of life and decrease menopausal symptoms [12].”
Comments: Methods
Participants: Zumba experience and frequency should be reported here.
Response: Added
Line 73 - 75: What were the classification ranges for body fat categories? And who/what were their origin?
Response: We added the categories defined by the World Health Organisation (WHO) based on BMI. We measured body fat too but did not use if for the classification.
Baseline Testing: No rationale was given for fitness testing? Were these used to characterize the physical fitness/capacity of participants? If so, this is highly unclear. This needs to be explained in the Methods as well as discussed in the Discussion. Without a rationale for these tests, the reader does not understand the reason, relevance, and implications of them.
Response: Thank you for your comment, we have added a large paragraph to justify these in the introduction, and also made it clearer here, we do hope that it helps.
Introduction:
“The fact that leaner women seem to spend more time in intense HR zones than women with greater body fat is very important to highlight because one of the main marketing strategies of the Zumba® Fitness industry is around weight loss. It is crucial to inform women that, based on the results previously described [13], they may not be able to work as hard, and hence lose as much weight as they first believed [14] if they are overweight, for example. It should be noted, however, that these results [13] need to be confirmed by further studies. Similarly, there might be other determinants of HR responses during Zumba Gold® classes that could be important to highlight to consumers. For example, older adults with a better aerobic capacity have been shown to reach significantly greater HR levels during physical exercise [15]. In addition, it was suggested that individuals with reduced balance may not be able reach high levels of HR during dance-based exercise sessions, because some dance steps (in particular backward steps) challenged their balance [16].”
In the methods, we just added our aim linked to the determinants of HR responses before describing he fitness tests.
“The following fitness characteristics were measured to investigate if they were determinants of HR responses during the classes:”
Line 87 - 88: Were pre-test procedures followed for BIA testing? If so, state them. Additionally, muscle mass was recorded; that should be stated here as well.
Response: Muscle mass was added. The pre-test procedures followed were: “Participants were required to avoid consuming caffeine 24 h before the measurements and to not perform any vigorous activity 8h before the measurements.” We could not consider any other pre-test procedure, such as no eating or drinking 4h before as the measurements took place at 10am (we didn’t want participants fasted while exercising).
Line 108: Define what the criteria used for a "qualified" instructor.
Response: added.
Line 108 - 109: Were all subjects equally familiar or unfamiliar with the songs, moves, etc. used in the sessions? Were song speeds similar? Were sessions, songs, moves of similar intensity?
Response: we added the following in the text to answer your questions:
“Each Zumba Gold® class was 60-min long, always included the same songs and dance moves, to make sure participants were equally familiar with the class. Classes were led by a qualified instructor, certified by the Zumba® Fitness company , and performed indoors in dance studios. Before each song, the instructor demonstrated the steps slowly, giving time to participants to learn them. Each session included one or two warm-up and cool-down songs and the main body was based on steps from the following six dance styles commonly used in Zumba: merengue, cumbia, reggaeton, salsa, belly dancing and pop. These songs differ by their intensity and type of steps involved.”
Line 124: I do not believe ACSM uses the "Zone" terminology (i.e. Zone 1, Zone 2, etc.). If citing ACSM, use to cite %HRmax intensity classifications (i.e. light, moderate, high-intensity). If using "Zone" terminology, cite reference of where that comes from
Response
Thanks for your comments. We deleted the reference to “zones” throughout the manuscript and use “HR intensity category”, or reference to the intensity name instead, which is the correct classification for ACSM.
Comments: Results
Table 1:
- n for All and each weight category needs to be included.
Response: done
- Zumba experience and Weekly PA units; is this weekly h? "h" is insufficient.
Response: Zumba experience is in year, this was changed, “h” was replaced by “hours”.
- Up to Zone 2 row, text is bold, after text is not bold. Needs to be consistent.
Response: this was changed
- All acronyms must be defined under table.
Response: Done
Line 151: "However" needs a space before previous sentence.
Response: Done
Line 146 - 158: These data need to be depicted in table of figure.
Response: The data was moved to a Table (Table 2), and the text was kept simpler.
Line 154 - 156: "Groups" need to be clearly defined and stated (i.e. weight groups").
Response: Done
Line 178: Years of experience should be in Table 1.
Response: This was moved, thanks!
Discussion
The paper begins with discussing health issues related to menopause, however, these are not discussed after that. The Discussion should tie back to those opening comments and discuss the implications that these data have on post-menopausal women's health. The Discussion currently does discuss the significance, implications, or applications of the data. The authors should include a discussion on these items to justify the study and to highlight it's relevance.
Response: Thanks for this comment, it is very helpful. We have added some paragraphs (and references) to address these aspects. However, we do not want to draw too many conclusions regarding the implications of our findings on health, as it was not the aim of our paper. Our aim was to describe acute responses to Zumba Gold classes, and whilst reaching a greater HR in the class should lead to long term health improvements, it would need to be proven by an intervention study to be a strong conclusion. We do hope the added paragraphs and references improve the quality of the paper.They are highlighted in red in the text and copied below:
“The benefits of exercising at moderate intensity or higher are well-established. Indeed, moderate intensity exercise utilises and enhances skeletal muscles ability to oxidise fatty acids, improves mitochondrial density and leads to improvements in aerobic capacity [25]. Moderate to high intensity exercise develops cardiac hypertrophy thus improves stroke volume and cardiac output [26]. These adaptations should therefore influence menopausal women’s health in the long term by prevent or delaying cardiovascular disease and promotint weight loss or maintenance, which are known health issues associated with the menopause [1-2].”
“Nevertheless, low intensity exercise improves circulation, increases muscular capillary beds and oxygen delivery [28], which could also contribute to some health benefits around the menopause, such as increasing aerobic capacity.”
“With consideration of the recommendation from the ACSM to spend 30 minutes at moderate intensity exercise per day alongside this study’s findings both aerobic fitness and fatty acid oxidation capacity can be improved through Zumba Gold and therefore help to mitigate some physiological effects of the menopause. With adaptations to the fat metabolism this aids weight management and improves cardiovascular health, therefore, the moderate or moderate to high intensity zones are beneficial for menopausal women.”